# RepViT-MedSAM: Efficient Segment Anything in the Medical Images

Qasim Ali[1][0009−0008−0630−0051], Yuhao Chen[1][0000−0001−6094−0545], and Alex Wong[1][0000−0002−5295−2797]

University of Waterloo, Waterloo, Ontario, Canada
{ m45ali, yuhao.chen1, alexander.wong }@uwaterloo.ca

**Abstract.** Segmenting medical images to identify lesions, organs and other areas of interest is crucial for diagnosis and treatment decisions. Traditionally, segmentation is accomplished through manual tools or using automated task-specific neural network models. A promising alternative solution to this problem is to create general-purpose models for segment anything in medical images, such as MedSAM [8]. These foundation models can segment regions across a multitude of modalities, at levels comparable to task-specific models. However, these models are often large and computationally expensive, preventing them from being used in clinical settings where they lack dedicated GPUs. We propose an efficient model for the segment anything in medical images problem, RepViT-MedSAM, created from a two step training process. First, the image encoder of MedSAM is distilled into a more efficient RepViT feature detector using aggressively augmented medical images. Then the entire end-to-end model, with the prompt encoder and mask decoder, is fine-tuned using ground truth masks and MedSAM's predictions. On the test set, RepViT-MedSAM surpasses the performance of baseline MedSAM in performance and efficiency, achieving an average Dice Similarity Coefficient (DSC) of 0.8528, an average Normalized Surface Distance (NSD) of 0.8666, taking a total execution time of 195 seconds, and ranking 12/23 among other contestants. RepViT-SAM offers a promising solution for real-world medical image segmentation with its efficiency and accuracy. The code for this project is available at https://github.com/icecap360/TurboMedSAM.

**Keywords:** Segment Anything · SAM · Medical Image Segmentation · Image Segmentation · Efficient Neural Networks · CNN · SAM · RepViT

## 1  Introduction

Medical imaging empowers physicians when making diagnostic and treatment decisions, giving them valuable insights into specific regions. Segmentation is a core component of medical imaging analysis, used to delineate regions of interest (ROI) a professional may deem useful [7]. Manual image segmentation is a labour-intensive and time-consuming process often requiring professional expertise. Many semi- or fully automatic segmentation algorithms are designed to

operate in particular image modalities or delineate particular anatomical structures and pathological regions [7]. These limitations indicate the need for a Foundation Model [2] for medical image segmentation, that can generalize to a wide variety of modalities and regions of interest.

Foundation Models, deep neural networks trained on broad data at scale, have been shown to generalize to a wide range of downstream tasks [2]. SAM [6] is a image segmentation foundation model used to delineate regions in natural images. However, SAM is a heavy transformer model with substantial computational requirements, preventing it from being used in resource-constrained environments. Mobile-SAM [16] creates a lightweight SAM for mobile hardware. They distill the image-encoder of SAM into a lightweight TinyViT, retaining the prompt encoder and mask decoder architecture. They train their model on a 1% of the SA-1B dataset [6]. Similarly, RepViT-SAM [11] follows an identical design to MobileSAM but use an efficient CNN (RepViT [10]) architecture for their image encoder. Efficient-ViT SAM [17] distills the image-encoder of SAM into an Efficient-ViT, retaining the prompt encoder and mask decoder. They train the image encoder on the whole SA-1B dataset [6].

MedSAM [8] is a recent foundation model for medical segmentation on a multitude of different modalities and regions of interest. It exhibits better accuracy and robustness than modality-wise specialist models. MedSAM is however a large model, with over 90 Million parameters, preventing it from being used in resource-constrained settings. This prevents MedSAM from being used in many clinical settings, where PCs often lack dedicated GPU hardware. The original authors released an efficient MedSAM, LiteMedSAM [1]. LiteMedSAM distills the ViT-B [4] image encoder of MedSAM into a TinyViT [13]. However, LiteMedSAM results in a significant decrease in performance relative to MedSAM.

This paper describes a solution of the "waterlooviplab" team to the "CVPR2024 Segment Anything in Medical Images on Laptop" challenge. The challenge involves training an efficient segment anything foundation model for medical imaging. A key constraint is that the evaluation platform consists of a 3.6GHz Intel Xeon CPU, 8G of RAM, and no GPUs. Another difficulty is the scale of the dataset, consisting of over 1.5 million training samples across 11 image modalities, which is large relative to the compute available to the team.

Building upon RepViT-SAM[11] architecture, our solution uses the RepViT[10] architecture as the image encoder instead of the heavy vision transformer (ViT-B) [4]. The first stage of training consists of distilling the MedSAM encoder into a RepViT backbone using aggressively augmented images. Afterwards, we fine-tune the full MedSAM model, with the prompt encoder and mask decoder using labeled ground truth (GT) masks and MedSAM[8] predictions. The final trained model surpasses MedSAM in performance and efficiency. On the test set, RepViT-MedSAM achieves an average Dice Similarity Coefficient (DSC) of 0.8528, an average Normalized Surface Distance (NSD) of 0.8666, a total execution time of 195 seconds, and ranking 12/23 among other contestants.

## 2    Method

Our proposed architecture follows the design described in SAM, but we replace the image encoder with the efficient feature detector RepViT. Our two step training procedure, distillation and fine-tuning, are illustrated in Figure 1. The usage of two phases helps to stabilize training between the encoder and decoder.

### 2.1   Preprocessing

We maintain the file format provided, saving each image-mask pair in the same npz file. We split 3D voxel grids into 2D slices. In the interest of simplicity, we use a PyTorch dataloader that loads image-mask pairs from the npz files. The bounding box prompts of the training set were computed on the fly using randomly perturbed ground truth masks. To preserve fine image details crucial for accurate analysis, bounding boxes, masks, and images are resized to 1024x1024. The input image is also normalized to a mean of 0 and standard deviation of 1. The mean and standard deviation were calculated from images on the training set.

### 2.2   Architecture

The architecture was chosen to be similar to the original MedSAM, but with a RepViT as the image encoder. The prompt encoder and mask decoder remain the same.

RepViT [10] is a MobileNetV3[5] based CNN that incorporates architectural design efficiencies of modern lightweight ViTs. It outperformed existing state of the art lightweight ViTs of similar size, while exhibiting favourable latency on mobile devices such as the iPhone12. We choose the RepViT-M1.1 variant, as its speed on CPU was significantly less than that of the contest baseline which used a TinyViT[13] backbone. Following [13], we set the stride of the last down-sampling depth-wise convolution to 1 (instead of 2) to make the output resolution compatible with the mask decoder in the original MedSAM.

Our main strategy for improving inference on CPU was to choose a CNN, RepViT, as our image encoder. The logic being that because CNNs contain many local operations and lack expensive global operators (such as self-attention) they would be more efficient on CPUs.

### 2.3   Training Pipeline

In the distillation phase of training 1a, the ViT-B image encoder of the MedSAM model is distilled into a smaller RepViT. Each image is augmented with random crop, random horizontal flip, and random vertical flip. Prior work showed that a student model can outperform its teacher provided that it has a sufficiently large model capacity and aggressive data augmentation is applied [3]. An aggressive augmentation can be described as one that reduces the covariance of the

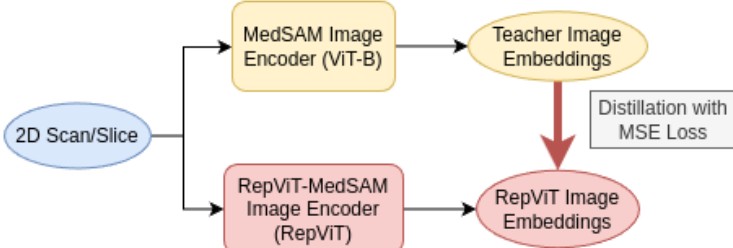

(a) Distillation process, MedSAM's image encoder is distilled using mean square error.

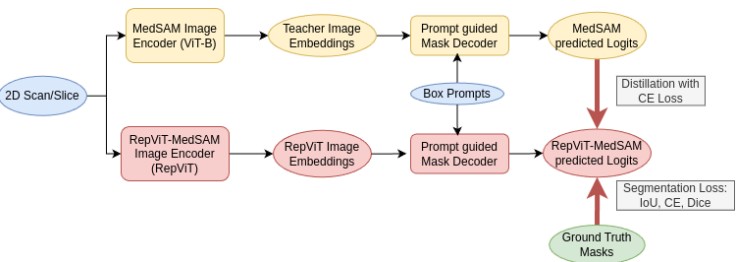

(b) Fine-tuning process, the loss is composed of two components, segmentation loss from the ground truth segmentations and distillation loss from MedSAM's image encoder.

Fig. 1: Illustrations of the two phases of training, Distillation and Fine-tuning. Blue ovals indicate data and rounded rectangles indicate models. Gray rectangles and their corresponding red arrows indicate losses.

teacher-student cross-entropy [12]. Given these findings, we choose to randomly apply one of the following three augmentations on a given batch - CutMix [15], MixPatch [3], or no augmentation. The loss function is simply the mean-square error (MSE) loss between the student feature map and the teacher feature map.

During the second phase of training, fine-tuning 1b, we train end to end with the prompt encoder and mask decoder. The prompt encoder and mask decoder are initialized with the MedSAM weights. Each image is augmented with basic transformations; random horizontal flip and random vertical flip. The loss during this stage contains two components, segmentation loss with the ground truth masks, and distillation loss where the teacher targets are the MedSAM predicted logits. The segmentation loss is designed to be well-suited for diverse medical image applications, composed of intersection over union (IoU) loss, cross-entropy (CE) loss, and dice loss [9]. The distillation loss is the cross-entropy between the predicted logits and MedSAM logits.

### 2.4   Post-processing

The final masks are computed by resizing the predicted logits to the original image size, followed by a sigmoid activation function.

## 3    Experiments

### 3.1   Dataset and evaluation measures

The dataset consisted of 1,516,396 image mask pairs from 11 image modalities, containing both 3D and 2D data. The dataset was significantly imbalanced, with CT scans and MR scans accounting for 78% and 12% of the data respectively. The CT and MR scans contain many more data samples primarily because the preprocessing decided to split the 3D voxels into 2D slices. The 2D image modalities exhibited a imbalance in their distribution, as the PET, Endoscopy, and X-Ray datasets take approximately 43%, 29%, and 22% respectively. We do not use any external public datasets for model development.

The evaluation metrics mirrored the ranking metrics used by the CVPR 2024 MedSAM on Laptop Contest. They included two accuracy measures—Dice Similarity Coefficient (DSC) and Normalized Surface Dice (NSD)—alongside one efficiency measure—running time.

### 3.2   Implementation details

**Environment settings** The development environments and requirements are presented in Table 1.

Table 1: Development environments and requirements. (mandatory table)

| System | Ubuntu 18.04.5 LTS |
|---|---|
| CPU | AMD Ryzen Threadripper PRO 5955WX 16-Cores 1800MHz |
| RAM | 264GB |
| GPU (number and type) | 3 NVIDIA RTX A6000 49G |
| CUDA version | 12.0 |
| Programming language | Python 3.11 |
| Deep learning framework | torch 2.4.0, torchvision 0.19.0, timm 0.9.16 |
| Specific dependencies | |
| Code | https://github.com/icecap360/TurboMedSAM |

**Training protocols** In the distillation phase of training, aggressive augmentation is used. Each image is augmented with random crop, random horizontal flip, and random vertical flip. Furthermore, we randomly apply one of the following three augmentations on all batches - CutMix [15], MixPatch [3], or no augmentation. During fine-tuning, we apply minimal data augmentation using just random horizontal flip and random vertical flip. We found that this prevented the corruption of the masks and improved training stability.

A class-balanced sampler was used, ensuring that the model performs equally well on all modalities regardless of the number of training images for each modality.

Three metrics are used to assess the performance of a model; Dice Similarity Coefficient (DSC), Normalized Surface Dice (NSD), and running time (sec). A model is better than another model if its performance on two of the three metrics is superior.

The training protocols during distillation are presented in Table 3 and protocols during fine-tuning are presented in Table 2.

## 4   Results and discussion

### 4.1   Quantitative results on validation set

The results of the proposed method on the validation set are displayed in Table 4. Our model outperforms MedSAM on both accuracy metrics, achieving an Average Dice Similarity Coefficient (DSC) of 0.8626 and Average Normalized Surface Distance (NSD) of 0.8828. This represents a significant improvement over MedSAM's 0.8528 DSC and 0.8666 NSD. As per the validation rankings it placed 6th in average DSC and 4th in average NSD, where its performance relative to the 5 higher ranked (as per DSC) teams lagged most severely in Ultrasound and X-Ray imagery.

The model performed best on Dermoscopy and Endoscopy based on average DSC and average NSD scores respectively. It performed most poorly on the PET and X-Ray modalities based on DSC. The results on the different modalities are expected, as Dermoscopy and Endoscopy are RGB images where the target is often large, clearly delineated and isolated in the frame.

Table 2: Training protocols during fine-tuning.

| | |
|---|---|
| Teacher Model | MedSAM [8] |
| Initial decoder and prompt encoder weights | MedSAM [8] |
| Batch size | 10 |
| Image size | 1024×1024×3 |
| Patch size | 16×16 |
| Total epochs | 9 |
| Optimizer | AdamW |
| Initial learning rate (lr) | 1e-3×10×3/256 |
| Lr decay schedule | Decay 0.1 after 2 epochs |
| Warmup | Learning rate 1e-4 for first epoch |
| Training time | 96 hours |
| Loss function | Dice loss with GT+ IoU with GT + CE with GT + CE with teacher |
| Number of model parameters | 12.56M |
| Number of flops | 86.15 GLOPs |
| $CO_2$eq | 45.395 kg |

Table 3: Training protocols during distillation

| | |
|---|---|
| Teacher Model | MedSAM [8] |
| Batch size | 10 |
| Image size | 1024×1024×3 |
| Patch size | 16×16 |
| Total epochs | 9 |
| Optimizer | AdamW |
| Initial learning rate (lr) | 5e-4 |
| Lr decay schedule | None |
| Training time | 48 hours |
| Loss function | MSE |
| Number of model parameters | 8.5M |
| Number of flops | 82.6 GLOPs |
| $CO_2$eq | 6.47 kg |

Table 4: Quantitative evaluation results on the validation set. Baseline corresponds to the performance of MedSAM[8], Ablation 1 corresponds to not using distillation during finetuning and Ablation 2 corresponds to not using using aggressive augmentation during distillation. All results are in percentage (%)

| Target | Baseline | | Ablation 1 | | Ablation 2 | | Proposed Ep4 | | Proposed Ep9 | |
|---|---|---|---|---|---|---|---|---|---|---|
| | DSC | NSD | DSC | NSD | DSC | NSD | DSC | NSD | DSC | NSD |
| CT | 0.9266 | 0.9532 | 0.9083 | 0.9387 | 0.9106 | 0.9407 | 0.9215 | 0.9523 | 0.9239 | 0.9541 |
| MR | 0.9041 | 0.9395 | 0.8605 | 0.9072 | 0.852 | 0.9008 | 0.8727 | 0.9168 | 0.8769 | 0.9208 |
| PET | 0.6312 | 0.4818 | 0.7255 | 0.5638 | 0.7263 | 0.61 | 0.6934 | 0.5272 | 0.7381 | 0.6172 |
| US | 0.9192 | 0.9555 | 0.7831 | 0.8289 | 0.8109 | 0.8584 | 0.8086 | 0.8556 | 0.8015 | 0.8478 |
| X-Ray | 0.7828 | 0.8401 | 0.3785 | 0.3637 | 0.7052 | 0.7679 | 0.7563 | 0.8162 | 0.759 | 0.8184 |
| Dermotology | 0.9137 | 0.9281 | 0.8961 | 0.9116 | 0.9344 | 0.9498 | 0.941 | 0.9563 | 0.9441 | 0.9592 |
| Endoscopy | 0.9683 | 0.9886 | 0.9282 | 0.9583 | 0.9175 | 0.9496 | 0.9204 | 0.9488 | 0.9313 | 0.9615 |
| Fundus | 0.9501 | 0.9664 | 0.9367 | 0.9534 | 0.9178 | 0.9359 | 0.9255 | 0.9435 | 0.9214 | 0.9395 |
| Microscopy | 0.679 | 0.7465 | 0.8271 | 0.8845 | 0.8383 | 0.9016 | 0.8648 | 0.9255 | 0.867 | 0.927 |
| Average | 0.8528 | 0.8666 | 0.8049 | 0.8122 | 0.8459 | 0.8683 | 0.856 | 0.8714 | 0.8626 | 0.8828 |

We also perform two ablations, using the same training protocols in Tables 3, 2 but in the interest of training time we report results 4 after only 4 epochs of fine-tuning instead of 9. Ablation 1 corresponds to not adding distillation loss (calculated from MedSAMs predicted logits) during the fine-tuning stage. Ablation 2 corresponds to not using aggressive data augmentations during the distillation phase. As can be seen from the results, neither ablation performs as well as the proposed method after 4 epochs of training.

### 4.2   Qualitative results on validation set

Figure 2 displays some results on the validation set, for which ground truth was unavailable. Figure 2a shows multiple successful segmentations, showcasing that given a prompt the model can successfully recognize the correct object. However several issues persist. Firstly segmentations often fail to correctly delineate precise boundaries of the lesion or organ. Figure 2b showcases undersegmentation, where the model fails to delineate regions close to the edges of the organ. In Figure 2c the segmentation leaks beyond the intended lesion. The model also fails to delineate fine details, as seen in the delineation of teeth of Figure 2d. Secondly the shape of the segmentations are sometimes incoherent, with non-smooth boundaries, disjoint elements and holes. For example, the segmentation associated with the dark green prompt in Figure 2d features a non-smooth boundary, contains disjoint components and internal gaps.

### 4.3   Segmentation efficiency results on validation set

The running time of several test cases are shown in Table 5. The measurements were taken on an Intel Core i9-9920X 3.50 GHz CPU. The average segmentation time of the proposed method and baseline (LiteMedSAM) was 2.53 sec and 3.184 sec respectively.

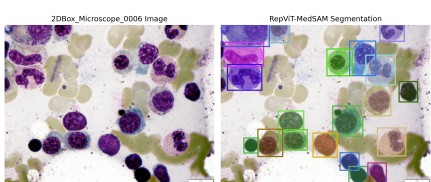

(a) Successful segmentation

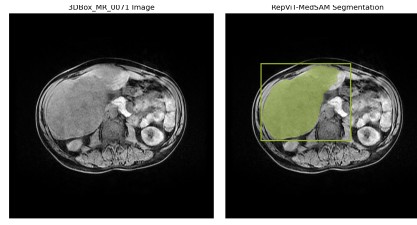

(b) Failure to delineate boundary of large isolated object.

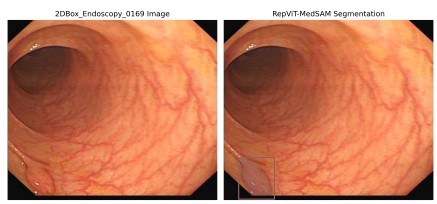

(c) Leakage of segmentation boundary.

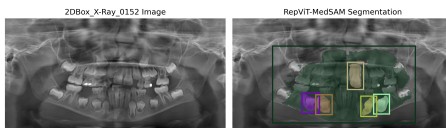

(d) Inability to delineate shape of small teeth, and incoherent segmentation of dark green prompt.

Fig. 2: Sample predictions of the proposed method.

Table 5: Quantitative evaluation of segmentation efficiency in terms of running time (sec).

| Case ID | Size | Num. Objects | Baseline | Proposed |
|---|---|---|---|---|
| 3DBox_CT_0566 | (287, 512, 512) | 6 | 197.977 | 161.506 |
| 3DBox_CT_0888 | (237, 512, 512) | 6 | 53.563 | 41.840 |
| 3DBox_CT_0860 | (246, 512, 512) | 1 | 7.6142 | 6.0781 |
| 3DBox_MR_0621 | (115, 400, 400) | 6 | 91.684 | 65.990 |
| 3DBox_MR_0121 | (64, 290, 320) | 6 | 53.810 | 43.988 |
| 3DBox_MR_0179 | (84, 512, 512) | 1 | 7.0516 | 5.8353 |
| 3DBox_PET_0001 | (264, 200, 200) | 1 | 4.5094 | 3.5300 |
| 2DBox_US_0525 | (256, 256, 3) | 1 | 0.3707 | 0.3072 |
| 2DBox_X-Ray_0053 | (320, 640, 3) | 34 | 0.9485 | 0.9216 |
| 2DBox_Dermoscopy_0003 | (3024, 4032, 3) | 1 | 0.6430 | 0.5223 |
| 2DBox_Endoscopy_0086 | (480, 560, 3) | 1 | 0.3644 | 0.3148 |
| 2DBox_Fundus_0003 | (2048, 2048, 3) | 1 | 0.4358 | 0.3652 |
| 2DBox_Microscope_0008 | (1536, 2040, 3) | 19 | 0.8528 | 0.7943 |
| 2DBox_Microscope_0016 | (1920, 2560, 3) | 241 | 6.2376 | 6.2868 |

### 4.4   Results on final testing set

The results of the testing set of the contest are shown in Table 6. The total execution time was 194.73 seconds, measured on an Intel Xeon W-2133 3.60 GHz CPU. The average DSC was 0.8528 and the average NSD was 0.8666. The submission ranked 12/23 submissions according to rank-then-aggregate strategy (i.e. average of rank across nine modalities and three metrics).

Table 6: Quantitative evaluation results on the test set. All results are in percentage (%)

| Target | DSC (%) | NSD (%) | Runtime (sec) |
|---|---|---|---|
| CT | 0.730759 | 0.786019 | 38.66128 |
| MR | 0.733583 | 0.693984 | 17.5917 |
| PET | 0.606909 | 0.544005 | 74.11131 |
| US | 0.887694 | 0.934459 | 9.666569 |
| X-Ray | 0.752008 | 0.855376 | 7.075892 |
| Endoscopy | 0.943635 | 0.970323 | 7.096959 |
| Fundus | 0.865523 | 0.885222 | 7.124286 |
| Microscopy | 0.881099 | 0.899042 | 7.176372 |
| OCT | 0.759746 | 0.823903 | 6.761073 |
| Average | 0.8528 | 0.8666 | 19.4739 |

### 4.5   Limitation and future work

While the proposed method is able to identify regions of various sizes, it struggles to form coherent segmentations and delineate accurate boundaries. The issues are particularly pronounced on objects with intricate shapes and objects in cluttered settings. A possible reason for these shortcomings is the reliance solely on high level feature maps. Incorporating low level feature maps that capture finer and more detailed semantics would help to alleviate this issue.

Further potential improvements include incorporating the 3D structure of voxel grids for 3D images, as treating each slice independently ignores the 3D relationships among the slices. Another potential improvement can be made to the final inference model, where pruning and quantization can be applied to improve inference speed.

## 5   Conclusion

In this paper we detailed the design of RepViT-MedSAM, an efficient model for the segment anything in medical images problem. We replace the image encoder of the MedSAM baseline with a RepViT variant. Our novel distillation pipeline consists of first aggressively distilling the image encoder features into

RepViT, and then fine-tuning the end to end model using ground truth masks and MedSAM's predicted logits. The model achieved an impressive average DSC of 0.8626 and average NSD of 0.8828. However key limitations remain, including the ability to delineate intricate boundaries and form well-defined segmentations.

**Acknowledgements** We thank all the data owners for making the medical images publicly available and CodaLab [14] for hosting the challenge platform.

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

Table 7: Checklist Table. Please fill out this checklist table in the answer column.

| Requirements | Answer |
|---|---|
| A meaningful title | Yes |
| The number of authors ($\leq$6) | 3 |
| Author affiliations and ORCID | Yes |
| Corresponding author email is presented | Yes |
| Validation scores are presented in the abstract | Yes |
| Introduction includes at least three parts: background, related work, and motivation | Yes |
| A pipeline/network figure is provided | Figure 1 |
| Pre-processing | 2.1 |
| Strategies to data augmentation | 3 |
| Strategies to improve model inference | 3 |
| Post-processing | 3 |
| Environment setting table is provided | Table 1 |
| Training protocol table is provided | Table 2, 3 |
| Ablation study | Table 4 |
| Efficiency evaluation results are provided | Table 5 |
| Visualized segmentation example is provided | Figure 2 |
| Limitation and future work are presented | Yes |
| Reference format is consistent. | Yes |
| Main text $>=$ 8 pages (not include references and appendix) | Yes |