# OpenReview forum: "RepViT-MedSAM: Efficient Segment Anything in the Medical Images"
_thecvf.com/CVPR/2024/Workshop/MedSAMonLaptop — CVPR24 MedSAMonLaptop_

### Official Review · Reviewer_nFNC · 2024-06-10
**Sufficient but the details still need to be described more clearly**

**Rating:** 7
**Confidence:** 5

**Review:**

This paper uses the distillation to distill the MedSAM into RepViT-MedSAM version, which is much efficient architecture than TinyViT of LiteMedSAM. The results look good with faster inference speed.

Comment:
The only concern of mine is that 9 epochs of distillation and training is sufficient?
What is the memory peak for distillation, as we need to load both models.

---

### Official Review · Reviewer_i8sw · 2024-06-13
**Very detailed experimental details are provided, and solutions are proposed for the problem of unbalanced data distribution, but there is a lack of thinking and discussion on the reasons for the success and failure of each mode.**

**Rating:** 7
**Confidence:** 3

**Review:**

The RepViT-MedSAM proposed in this article follows the design and experimental ideas of Mobile-SAM, and has been improved to address the problem of uneven distribution of multi-modal medical images, which is a highlight of the article.
Additional data augmentation is utilized in the model distillation stage, but there is a lack of discussion on the effects of using data augmentation and insufficient details on how to improve the problem of uneven data distribution.
Highlights:
1. Detailed experimental details are provided and the complete project is open sourced.
2. Improvements have been made to address the problem of uneven multi-modal data distribution.
3. The overall structure of the article is complete and provides sufficient details for reproduction.
Areas for improvement:
1. The problem of uneven distribution of multi-modal data quantity was discovered, and a unique sampling strategy was proposed, hoping to reveal more details and increase ablation experiment comparison.
For example, at a certain sampling ratio, the model can ultimately achieve a more balanced result.
2. There is insufficient discussion on data enhancement, why data enhancement strategies such as Mixup should be selected, and how much improvement can be achieved after selecting these strategies
3. The final model has been improved in some modes, but has declined in some modes. This can be discussed appropriately.
4. Open source projects can add README files to provide simple instructions for reproducing the project.

---

### Official Review · Reviewer_vctC · 2024-06-16
**A efficient and effective method**

**Rating:** 8
**Confidence:** 4

**Review:**

The author proposed an efficient model for segmentations in medical images, by knowledge distillation from MedSAM image encoder to Rep-ViT.

The manuscript is basically complete. Suggestions or deficiencies:
1. Efficiency description is ambiguous in the abstract.
2. The width between paragraphs is too wide on page 2.
3. Tables in **Experiments** section have no vertical line.
4. Too much space between Table 2 and Table 3.

---

### Decision · Program_Chairs · 2024-10-01

Accept